# Development of an occupational advice intervention for patients undergoing elective hip and knee replacement: a Delphi study

Paul Baker ,[1,2] Lucksy Kottam,[1] Carol Coole,[3] Avril Drummond,[4] Catriona McDaid ,[5] Amar Rangan ,[1,2,6] On behalf of the OPAL study group

For numbered affiliations see end of article.

**Correspondence to**
Paul Baker;
paul.baker1@nhs.net

## ABSTRACT

**Objective** To obtain consensus on the content and delivery of an occupational advice intervention for patients undergoing primary hip and knee replacement surgery. The primary targets for the intervention were (1) patients, carers and employers through the provision of individualised support and information about returning to work and (2) hospital orthopaedic teams through the development of a framework and materials to enable this support and information to be delivered.

**Design** Modified Delphi study as part of a wider intervention development study (The Occupational advice for Patients undergoing Arthroplasty of the Lower limb (OPAL) study: Health Technology Assessment Reference 15/28/02) (ISRCTN27426982).

**Setting** Five stakeholder groups (patients, employers, orthopaedic surgeons, general practitioners, allied health professionals and nurses) recruited from across the UK.

**Participants** Sixty-six participants.

**Methods** Statements for the Delphi process were developed relating to the content, format, delivery, timing and measurement of an occupational advice intervention. The statements were based on evidence gathered through the OPAL study that was processed using an intervention mapping framework. Intervention content was examined in round 1 and intervention format, delivery, timing and measurement were examined in round 2. In round 3, the developed intervention was presented to the stakeholder groups for comment.

**Consensus** For rounds 1 and 2, consensus was defined as 70% agreement or disagreement on a 4-point scale. Statements reaching consensus were ranked according to the distribution of responses to create a hierarchy of agreement. Round 3 comments were used to revise the final version of the developed occupational advice intervention.

**Results** Consensus was reached for 36 of 64 round 1 content statements (all agreement). In round 2, 13 questions were carried forward and an additional 81 statements were presented. Of these, 49 reached consensus (44 agreement/5 disagreement). Eleven respondents provided an appraisal of the intervention in round 3.

**Conclusions** The Delphi process informed the development of an occupational advice intervention as part of a wider intervention development study. Stakeholder

### Strengths and limitations of this study

► Wide stakeholder engagement including patients, employers, surgeons, general practitioners, allied health professionals with knowledge of the process of returning to work after hip or knee replacement.

► Recruitment of a large number of Delphi members who were invited to participate in all three Delphi rounds.

► Initial Delphi statements developed from research performed as part of the Occupational advice for Patients undergoing Arthroplasty of the Lower limb study (Health Technology Assessment 15/28/02).

► Loss of participants between each of the rounds of questioning, specifically the employer group who failed to participate in the final Delphi round.

► The process recruited participants from the UK only, which potentially limits the generalisability of findings to healthcare settings internationally.

agreement was achieved for a large number of intervention elements encompassing the content, format, delivery and timing of the intervention. The effectiveness and cost-effectiveness of the developed intervention will require evaluation in a randomised controlled trial.

**Trial registration number** International Standard Randomised Controlled Trials Number Trial ID: ISRCTN27426982

## INTRODUCTION

Hip and knee osteoarthritis are associated with a reduction in work participation and productivity and an increased risk of work loss.[1 2] The costs associated with occupational musculoskeletal disorders are significant.[3 4] The estimated annual cost of workplace ill health is £9.7 billion, equivalent to £18 400 per case.[5] These costs are borne not only by the individual (impact of ill health on quality of life), but also by their employers and society (loss of productivity, need for healthcare, rehabilitation and compensation).[3 4] In addition to the financial costs, lengthy sickness absence can

result in work disability, poorer general health, increased risk of mental health problems and higher mortality.[6–8] Working, therefore, has physical and mental health benefits, alongside its socioeconomic value.

Lower limb joint replacement is an effective and cost-effective treatment for patients with hip and knee osteoarthritis.[9–12] Recent changes to the pension age combined with an ageing UK workforce have resulted in a steady increase in the numbers of hip and knee replacements being performed in patients of working age over the last decade.[13–15] In 2017, 18 812 (20.5% of all hip replacements) and 17 765 (17.4% of all knee replacements) were performed in patients aged less than 60 years.[13–15] Current recommendations supporting return to work after hip and knee replacement are limited and inconsistent.[16] There is variation in the content, delivery and format of occupational advice delivered to patients having hip and knee replacements and a need to provided more comprehensive, individualised advice for these patients to support early, sustained return to work after surgery.[16]

The Occupational advice for Patients undergoing Arthroplasty of the Lower limb (OPAL study) was a National Institute for Health Research-Health Technology Assessment commissioned research study that aimed to develop an occupational advice intervention to support return to work after hip and knee replacement.[16] OPAL used an intervention mapping framework supported by related qualitative and quantitative work streams.[16] Initial research evaluated the specific needs of the population of patients who were in work and intended to return to work following surgery, established how individual patients returned to work and documented the barriers preventing return to work.[17 18] Through these work streams a range of key performance indicators and potential intervention components that could be used to develop an occupational advice intervention emerged.

To refine these components and address areas of uncertainty relating to the intervention a multistakeholder intervention development group was constructed to ascertain whether agreement could be reached about the design, content, delivery, format and timing of the proposed occupational advice intervention. To facilitate this process a modified Delphi consensus process was employed.[16] The Delphi approach was chosen as it can be delivered remotely in a short time frame without the need to convene meetings. It also enables researchers to collect the opinions of a range of different individuals with differing areas of expertise which was desirable in this setting. The initial research performed as part of the intervention mapping process provided the basis for this process by generating an initial list of statements for the Delphi consensus development.

## METHODS
### Design of the modified Delphi study
A modified 3 round Delphi consensus process was used.[19–21] The process was guided by the information

> **Box 1  Key intervention components identified during opal phase 1**
>
> ► Education and advice.
> ► Vocational counselling and guidance.
> ► Physical therapy and exercises.
> ► Work simulation/work hardening and job accommodation.
> ► Contact with employer/workplace visits.
> ► Multidisciplinary team involvement.

gathered from research completed during the first phase of the OPAL project.[16–18]

During the first phase of the OPAL project, a number of intervention components emerged that were considered likely to be integral to the development of a successful occupational advice intervention (box 1). Expanded versions of these components were used as the basis for initial statement development that could be explored during the Delphi process.

### Delphi stakeholder recruitment
Five stakeholder groups were identified for inclusion in the modified Delphi process. The sampling strategy for each stakeholder group is outlined in table 1, with participants chosen via a targeted approach to maximise patient, public and professional engagement. To ensure wide participation and the validity of the consensus process the process aimed to recruit a minimum of five individuals from each stakeholder group. A maximum limit of 15 individuals from any given stakeholder group was chosen to ensure one group's opinions did not overwhelm the opinions of others within the consensus process. As such, we aimed to have a minimum of 25 participants and a maximum of 75 participants for each round.

Although there are no definitive rules about the sample size for a Delphi study, a minimum of 8–10 participants has been suggested.[22] While higher response rates and ease of administration are an advantage of smaller homogeneous groups, we considered a larger sample size desirable given the variation in expertise and the heterogeneity within our stakeholder groups. Furthermore, if areas of uncertainty are being explored larger sample sizes can help to reduce errors and improves the reliability of the findings.[23]

Prior to enrolment, potential participants from all stakeholder groups were invited to participate via an email from a member of the OPAL study team as per the sampling strategy for each stakeholder group outlined in table 1. This email included a participant information sheet describing the Delphi consensus process and what participation involved. Participants were asked to confirm their consent to participate by return of email and only those that responded indicating their willingness to participate were included in the process.

### Development of Delphi statements
Prior to commencing, statements relating to the proposed content, format, delivery, timing and measurement of an

**Table 1** Sampling strategy used to identify Delphi members

| Stakeholder group | Requirement for inclusion | Participants recruited via |
| --- | --- | --- |
| Patients | Experience of returning to work after hip or knee replacement in the previous 12 months. | ▶ National Joint Registry patient network.<br>▶ British Orthopaedic Association patient group.<br>▶ Patient participants from OPAL phase 1. |
| Employers and occupation health services | Experience of managing an employee returning to work after hip or knee replacement in the previous 12 months. | ▶ Federation of Small Businesses.<br>▶ Make UK —The manufacturers organisation.<br>▶ Confederation of British Industry.<br>▶ Trade Union Congress.<br>▶ Department for Work and Pensions.<br>▶ The Fit for Work Service.<br>▶ The Work Foundation.<br>▶ The Society of Occupational Medicine.<br>▶ Institution of Occupational Safety and Health.<br>▶ Society of Occupational Health Nurses. |
| Orthopaedic surgeons | Surgeons undertaking a minimum of 20 hip or knee replacements per year. | ▶ British Hip Society.<br>▶ British Association for Surgeon of the Knee.<br>▶ British Orthopaedic Association.<br>▶ National Joint Registry. |
| Allied Health Professionals (AHPs—Physiotherapists and Occupational therapists) and nurses | AHPs actively involved in the assessment and/or management of patients undergoing hip or knee replacement. | ▶ Association of Chartered Physiotherapists in Occupational Health and Ergonomics.<br>▶ Chartered Society of Physiotherapy.<br>▶ Occupational therapy networks, for example, College of Occupational Therapists Specialist Sections in Work and Trauma and Orthopaedics.<br>▶ Royal College of Nursing. |
| General practitioners | Experience of managing a patient returning to work after hip or knee replacement in the previous 12 months. | ▶ Local Medical Committees.<br>▶ Royal College of General Practitioners.<br>▶ Local Clinical Commissioning Groups. |

OPAL, Occupational advice for Patients undergoing Arthroplasty of the Lower limb.

occupational advice intervention were developed. Due to the breadth of statements developed and their inter-related nature, we adopted a stepwise approach to the presentation of individual statements to the Delphi group. Round 1 focused on defining the content of the intervention in two sections. Section 1, focused on passive content ('written' advice and information) and section 2 on active content (actions or processes for patients, employers and healthcare members to undertake). These statements were piloted by a small sample of surgeons, general practitioners (GPs) and patients. Having first defined the content, we then used this information to refine the statements relating to the format, delivery, timing and measurement of this content presented in round 2. In round 2, statements were grouped under headings allowing exploration around specific themes. Round 3 was then used to clarify any areas of residual uncertainty from rounds 1 and 2 and present the proposed occupational advice intervention back to the Delphi participants for final comments.

For each statement within the Delphi process, participants were asked to rate the extent of agreement with individual statements about the importance of including specific elements in a occupational advice intervention, with possible options being: strongly agree; agree; disagree; strongly disagree; do not know. For a subset of statements in round 1, they were also asked to rate the deliverability of the content or action alongside current healthcare provision. Therefore, for some statements the participants were asked to provide two ratings one for 'importance' and one for 'deliverability'.

At the end of each section, there was a free-text box where participants could add suggestions relating to the intervention that could be evaluated in subsequent rounds. In rounds where statements from a previous Delphi round were being represented, these were presented alongside controlled feedback with modal round one rating for these statements; the proportion of each response option selected by the other participants; and a reminder of the participant's own previous ratings.

### Delivery of Delphi survey

The Delphi survey was delivered via email using an online web-based survey platform.[24] Round 1 was delivered between 25 September 2017 and 13 October 2017, round 2 between 22 November 2017 and 13 December 2017 and round 3 between 1 June 2018 and 22 June 2018. The email included a covering letter to the participants and an electronic link to the questionnaires. All three rounds allowed a minimum of 3 weeks for participants to respond. Automated reminders were sent via the electronic system after 10 days from the day of initialising the

survey. A further personalised email reminder was sent to non-responders during the final week of the surveys.

Round 1 and 2 questionnaires required respondents to provide their initials and occupation. All round 2 emails incorporated an overall report summarising the pooled responses from round 1 survey and, where appropriate, the responses of each of the five stakeholder groups. In addition, for those participants who completed the round 1 survey, an individualised report summarising their responses to the statements in round 1 were included with the round 2 survey to allow participants to reappraise their responses in view of the overall responses.[25] Round 3 emails included four core documents from the developed occupational advice intervention (a summary of the intervention, occupational checklist, patient 'return to work' workbook and employer booklet) for participants to review and comment. Email reminders were sent to non-responders during the final week of the surveys.

## Analysis of data

Descriptive analyses of the Delphi responses were undertaken by the OPAL study team. Results of each round were discussed with the wider OPAL study research team before the statements were agreed for subsequent rounds.

An a priori consensus threshold of 70% (strongly agree/agree or strongly disagree/disagree) was agreed before statements were circulated.[25] There is no universal agreement on an acceptable level of consensus for a Delphi study,[26 27] however, reports suggest this should be decided before commencing the study and recommends at threshold of at least 70% to ensure validity of the findings.[27] For statements that failed to reach consensus, further analysis was undertaken based on responses for each of the five stakeholder subgroups. The following rules were then employed to determine which statements were discarded and which were represented in the next round.

► If no or only one stakeholder group reached concordant consensus (≥70% agreement or disagreement) then the statement would be withdrawn.
► If two or more stakeholder groups reached concordant consensus (≥70% agreement or disagreement) then the statement would be represented in a subsequent round.
► In the situation where one or more stakeholder groups reach 'agreement' and another group reach 'disagreement' the statement would be discussed by the OPAL investigator team and a decision on inclusion/exclusion of the statement would be made.

For statements that were rated for both importance and deliverability in round 1, consensus was reached if the 70% threshold was achieved for both the importance and deliverability rating. Statements that reached consensus for one of the domains were analysed by stakeholder group as described above.

In rounds 1 and 2, statements reaching consensus were ranked according to the distribution of responses to create a hierarchy of agreement.

In round 3, the occupational advice intervention and associated documents were circulated for comment. Descriptive open feedback from participants to these documents were recorded.

## Patient and public involvement

The OPAL research project was developed in collaboration with members of the British Orthopaedic Association (BOA) Patient Liaison Group (PLG). A patient coapplicant from the BOA PLG was involved in the development of the research question and defining the outcome measures used within the wider OPAL study. Patients were involved in the design of the study from inception of the project, through protocol development, study delivery and project dissemination. These included patients from the BOA PLG, the National Joint Registry patient group and patient and public groups affiliated with the sponsor site.

## RESULTS
### Round 1

Responses were received from 43 of the 66 participants (65%) including 14 patients, 8 surgeons, 6 GPs, 11 allied health professionals and nurses, and 4 employers. In section 1 ('written' advice and information), consensus was reached for 26 of 32 statements (81%). Of the remaining six statements, five reached consensus for two or more stakeholder groups and were therefore taken forward to round 2 and one statement was discarded. Section 1 statements reaching consensus and ranked based on the strength of consensus are listed in table 2.

In section 2 (actions or processes for patients, employers and healthcare members to undertake), participants were asked to rate both the importance and deliverability of each statement. Of the 32 components presented, only 10 (31%) reached consensus for both importance and deliverability (table 3). Of the remaining 22 statements, 14 reached consensus for importance but not deliverability, 2 reached consensus for deliverability but not importance and 6 did not reach consensus for either. Of these statements seven reached consensus for both importance and deliverability for two or more stakeholder groups and were therefore taken forward to round 2 and 15 statements were discarded.

### Round 2

Responses were received from 26 of the 66 participants (39%) including 8 patients, 7 surgeons, 3 GPs, 6 allied health professionals and nurses, and 2 employers.

Twelve questions carried forward from round 1 plus one additional question generated from the free-text comments were presented to the participants. Of these, 10 reached consensus based on their potential importance within the proposed occupational advice intervention.

A further 81 statements grouped into 13 categories were then rated. This allowed the team to explore different approaches to a given problem. For example,

**Table 2** Statements descriptions reaching consensus for section 1 (ordered by % of respondents that strongly agreed or agreed)

| Is it important that an occupational advice intervention commenced prior to hip or knee replacement includes the following | Agreement (%) |
|---|---|
| Q9. Information about exercises and rehabilitation following surgery. | 100 |
| Q13. Information about returning to driving. | 100 |
| Q3. A broad overview written for all stakeholders, of what to expect following surgery (rates and timing of expected recovery). | 98 |
| Q15. Information about managing pain, types of analgesia and side effects. | 98 |
| Q5. Information about postoperative precautions, restrictions and activities to avoid following surgery. | 95 |
| Q18. Information about symptom management in relation to return to work and specific occupations, for example, expected levels of fatigue, pain, swelling. | 95 |
| Q12. Tips and tricks to help the patient manage around their home with day to day activities immediately following surgery. | 95 |
| Q10. Information regarding postoperative complications and their management. | 95 |
| Q14. Signposting to Driver and Vehicle Licensing Agency (DVLA) guidance. | 95 |
| Q23. Information for the patient about who to ask if they are having a problem returning to work. | 93 |
| Q4. Information about expected level of function at different time - points following surgery. | 88 |
| Q29. Advices about adaptions to working patterns to assist return including the use of phased returns, modified hours and altered work schedules. | 88 |
| Q21. Information and resources to support self-advocacy and empowerment. | 88 |
| Q20. Information about when it might be appropriate for patients and employers to access occupational health services. | 88 |
| Q19. Information for patients and employers about how to access occupational health services. | 88 |
| Q28. A list of potential workplace modifications, aids and adjustments that could be used to assist with return to work, with examples. | 84 |
| Q27. Information for the patients about how to ask for help at work from their employer and colleagues. | 84 |
| Q31. Guidance on how to set an appropriate provisional return to work date based on the date and type of surgery. | 81 |
| Q16. Guidance for orthopaedic care teams and GPs on how to use and prescribe a fit note. | 81 |
| Q24. Signposts to national and local support services for example, Fit4Work, citizens advices, Advisory, Consiliation and Arbitration Service (ACAS). | 81 |
| Q11. Information about how having surgery may impact on social relationships. | 81 |
| Q32. Advice about how psychosocial and emotional factor influence return to work. | 79 |
| Q6. Information about how long the hip and knee replacement prostheses will last. | 77 |
| Q17. Examples of the correct use of fit notes. | 77 |
| Q30. A list of potential return to work barriers for patients and employers to consider. | 74 |
| Q8. Information about managing more than one joint replacement in close succession. | 72 |

GPs, general practitioners.

the first category asked participants to rate a set of five statements relating to which healthcare team member should have responsibility for delivery and coordination of the occupational advice intervention. If at least one or more statements in a given category reached consensus this was taken as representative of the Delphi group's position relating to the given category and the remaining statements were discarded. Overall 49 statements (60%) reached consensus (44 agreement and 5 disagreement), at least one statement in every category reached consensus (online supplementary appendix table 1).

### The occupational advice intervention

Based on the evidence gathered throughout the OPAL study and consolidated through Delphi rounds 1 and 2 the occupational advice intervention was further developed and finalised.

The intervention was designed to support patients throughout their surgical pathway, starting during their initial outpatient appointment and continuing until 16 weeks after surgery. It had a number of key themes that linked to performance objectives for patients and staff and was supported by a range of patient and staff

**Table 3** Statements descriptions reaching consensus for section 2 (ordered by % of respondents that strongly agreed or agreed)

| How important/deliverable do you believe the following components are if an occupational advice intervention commencing prior to hip or knee replacement were to be developed | Agreement (%) | Agreement (%) |
|---|---|---|
| **Ten statements reaching consensus for both importance and deliverability** | | |
| Q37. A postoperative mechanism for the identification of patients that are not progressing toward return to work as planned. | 95 | 71 |
| Q52. Guidance for health services defining 'best practice' for patients returning to work after hip and knee replacement surgery. | 93 | 82 |
| Q45. Training for members of the hospital orthopaedic care team to increase awareness about return to work issues. | 88 | 82 |
| Q42. Interaction between the healthcare team and patient by phone, email or 'on-line' so that members of the care team can monitor progress and help the patient use the advice and information provided. | 88 | 70 |
| Q64. Guidance on when in the return to work process patients can safely be discharged back to primary care for continued management of their return to work. | 86 | 80 |
| Q36. A mechanism for preoperative identification of patients at 'high risk' of prolonged sickness absence following surgery. | 86 | 74 |
| Q51. Routine preoperative therapy assessment during which a return to work plan is developed between the patients and the hospital orthopaedic care team. | 84 | 80 |
| Q40. A separate intervention for hip and knee replacement patients that are not progressing towards return to work as planned. | 84 | 79 |
| Q62. A process by which work status can be included in referral information for all patients referred from primary care into secondary care for consideration of hip or knee replacement. | 79 | 79 |
| Q57. Information from patients that have experienced the process of returning to work after hip or knee replacement within the preoperative education process. | 76 | 73 |

resources (to support delivery and measurement of the intervention). A simplified schematic of the OPAL 'return to work' intervention is presented in figure 1.

### Round 3

In round 3, the finalised occupational advice intervention along with selected patient and staff materials were circulated to 65 of the 66 Delphi participants for comment (one patient withdrew). Responses were received from 11 participants comprising a constructive appraisal of the intervention from 9, as well as highlighting typographical and formatting issues. The feedback was positive in all cases.

A diagram of the overall Delphi consensus process is shown in figure 2.

### DISCUSSION

The Delphi consensus methodology was used to underpin the development of an evidence-based, theory-driven occupational advice intervention to assist patients returning to work after elective hip and knee replacement. It enabled the OPAL study team to rationalise the content, format, delivery and timing of the intervention and clarified areas of uncertainty related to the intervention that had arisen during the earlier stages of the research. Response to the developed intervention during

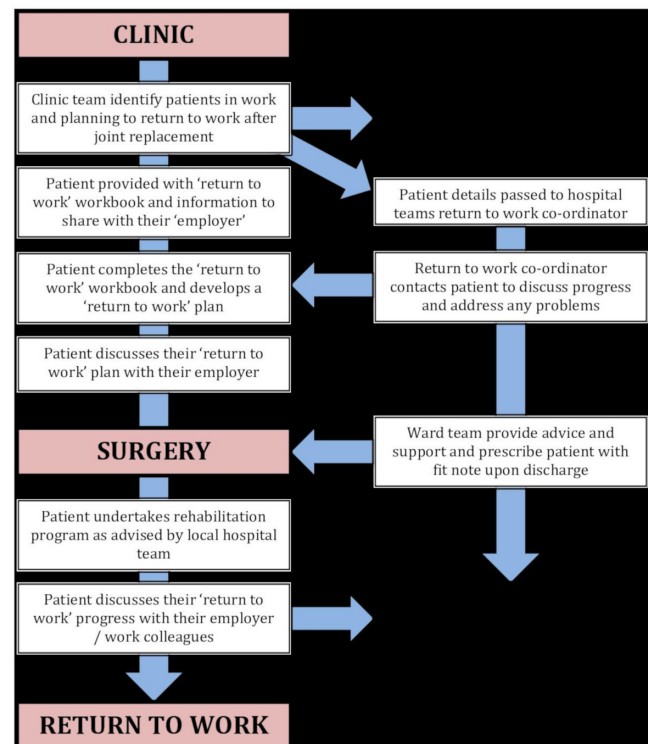

**Figure 1** Simplified schematic of the OPAL 'return to work' intervention.

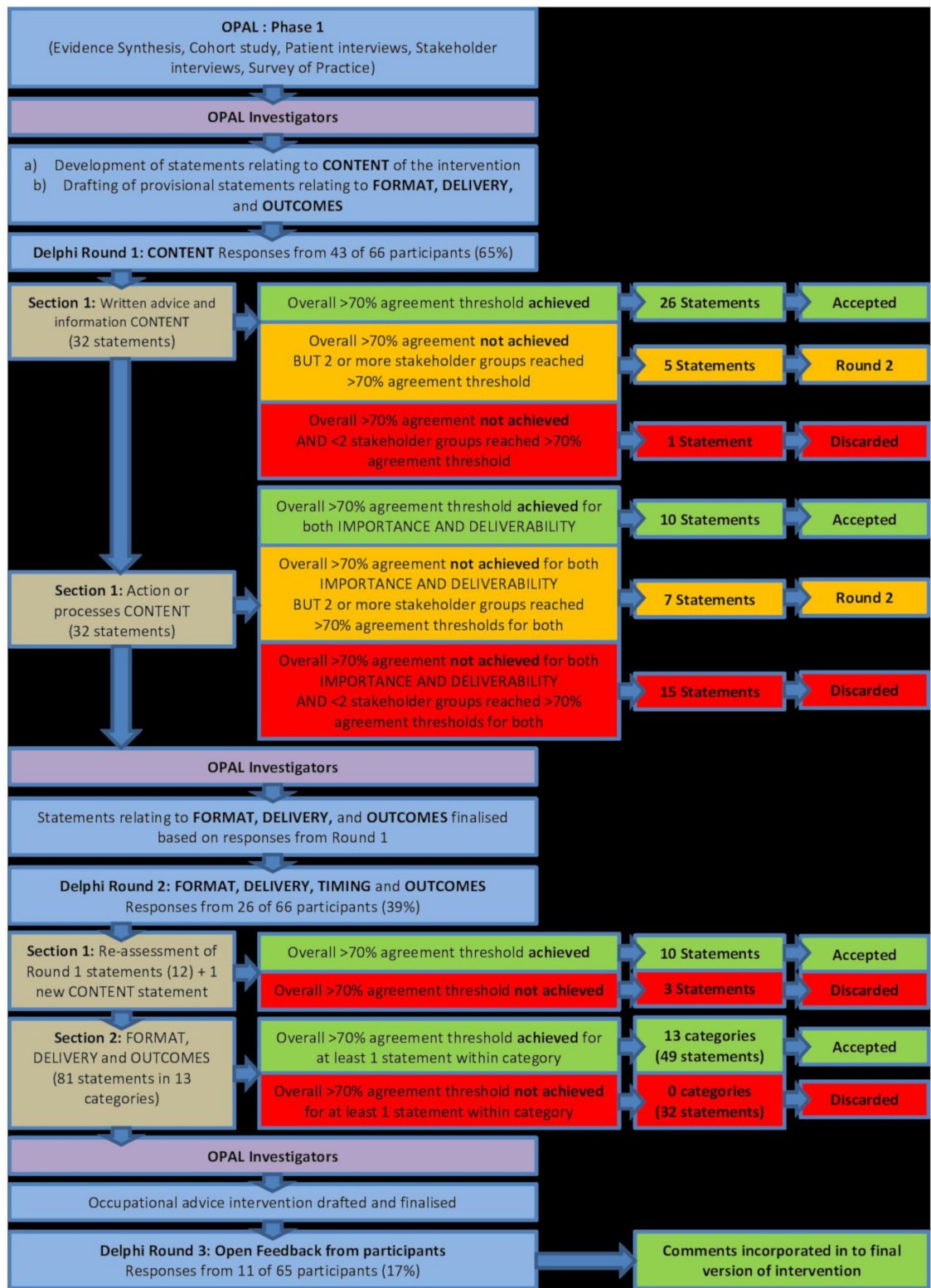

**Figure 2** Overview of the opal Delphi process. OPAL, Occupational advice for Patients undergoing Arthroplasty of the Lower limb.

the third round of the Delphi process was positive, validating the use of the Delphi process to support intervention development.

Prior to the Delphi process, the OPAL study had already completed a number of complementary research

phases to enable the OPAL team understand the current evidence, stakeholder and patient perspectives, and current practice relating to return to work after hip and knee replacement.[16–18] Through the intervention mapping framework this is information generated a

range of components for our intervention. The Delphi methodology was then used to 'refine' the intervention and reach consensus on the final design. This is similar to the modified Delphi approach used by Vonk Noordegraaf *et al* to develop a return to work intervention for gynaecological surgery, as it used existing evidence as the basis for the process but sought to bridge gaps and clarify uncertainty within this evidence.[28] However, one limitation of this approach is that it may inadvertently narrow the focus of the intervention with only components deemed important by the research team included. There is a risk that potentially useful intervention components that may have been of interest to the stakeholder groups were not included as the starting position was predefined. However, the approach used is not unusual and is similar to the approaches used by others.[20 29] Furthermore, given the breadth of work completed earlier in the OPAL study and the design of the modified Delphi survey allowing participants to suggest new intervention components within each round, this is unlikely to have had a negative impact.

Broad stakeholder involvement helped the research team ensure the final intervention was acceptable to all groups, increasing the chances of success when implemented and delivered. Unfortunately, despite good initial engagement, the response rates reduced as the process progressed. This is a common finding during Delphi processes[20] and was perhaps related to the larger sample size involved and extended period of the process with a 6-week gap between rounds 1 and 2 and a 6-month gap between rounds 2 and 3. The gap between rounds 2 and 3 was necessary as the intervention needed to be finalised with associated materials being developed during this period. Other contributing factors may include the increasing length of the Delphi questionnaires with each round and the volume of materials that needed to be reviewed in round 3. All participants were UK based and working with the setting of the UK National Health Service and social care provision or UK employment. Therefore, this may impact on the generalisability of the findings outside of the UK health setting.

While the low response rate in round 3 may be a concern, the purpose of this round was to circulate and draw comment regarding the final intervention rather than reach consensus on specific points. With 11 respondents, including at least one member from each stakeholder group, this seems valid as Delphi process relies more on the group dynamics even with reaching consensus rather than their statistical power and a lower limit of 10 participants is often considered sufficient for a Delphi panel.[30 31] During the process, there was a notable drop off in employer respondents. In total 12 employers initially expressed an interest in participating, however, only four responded in round 1, two in round 2 and one in round 3. It is often difficult to engage employers in research[32] and despite using a number of complementary strategies (17) we were unable to maintain engagement. However, as the intervention was designed to be delivered

in secondary care rather than in the workplace this potentially did not significantly influence the nature of the final intervention.

The modified Delphi methodology employed in this study resolved uncertainties about a number of intervention components. However, there were a few areas where the consensus process was limited. Two key areas that stakeholders felt were important were (1) the provision of additional pre and post-operative physiotherapy/occupational therapy (over and above standard care) in which return to work issues could be addressed and (2) the identification of 'high-risk' patients that should be provided with additional help and support. Yet, these positions conflicted with other information gathered from the Delphi participants and the evidence from OPAL phase 1. Essentially first, our cohort study failed to identify a 'high-risk' population and the current literature describing predictors of return to work after hip and knee replacement was limited.[33–38] This meant we were not able to confidently identify a 'high-risk' group in need of a more intensive targeted intervention. Second, there was concern about the cost, time and logistics associated with the implementation of a resource intensive intervention requiring additional patient interactions. The survey of practice and stakeholder/patient interviews demonstrated that services varied significantly in their structure and the resources available.[18] To be successful it was agreed that the intervention should supplement rather than replace existing pathways of care and should, where possible, use existing staff and adapt current working. Comments from Delphi participants, the OPAL research team and the study steering committee had similarly raised concerns about the implementation and sustainability of an intervention requiring significant additional resources. The OPAL research team therefore felt that, despite this component reaching consensus, it was prudent to pursue a less intensive model to improve implementation of the final intervention.

We were unable to compare our intervention to other occupational advice interventions for patients undergoing hip and knee replacement as no such interventions have been reported in the literature. A rapid evidence synthesis performed earlier in the OPAL study (PROSPERO protocol registration number CRD42016045235) found only four studies that reported occupational advice interventions for patients undergoing elective surgery. This included two randomised clinical trials (RCTs) from Belgium and the Netherlands in patients undergoing gynaecological surgery and lumbar disc surgery[39 40] and two qualitative studies that explored factors affecting return to work from the perspective of the patient following knee replacement[41] and factors influencing work disability following mastectomy.[42] Of the two interventions described in the RCTs one described a personalised e-Health intervention[39] whereas the other assessed a rehabilitation-orientated intervention focusing on early resumption of activities.[40] Our intervention drew on elements of both of these interventions in terms of

delivering an individualised patient-centred approach while encouraging early resumption of workplace activities through discussion with employers alongside workplace adaptions and alterations to working patterns.

In conclusion, a modified Delphi consensus process employed within a wider intervention development project facilitated the development of the OPAL occupational advice intervention. Consensus was reached for a range of intervention components that allowed the content, format, delivery and timing of the intervention to be finalised. The intervention developed and the materials created to support its delivery were well received by the Delphi group. The effectiveness and cost-effectiveness of the developed intervention will require evaluation in an RCT.

**Author affiliations**
[1]Department of Orthopaedics, South Tees Hospitals NHS Foundation Trust, Middlesbrough, UK
[2]Department of Health Sciences, University of York, York, UK
[3]School of Health Sciences, Faculty of Medicine & Health Sciences, University of Nottingham, Nottingham, UK
[4]Division of Rehabilitation, Ageing and Wellbeing, University of Nottingham, Nottingham, UK
[5]York Trials Unit, University of York, York, UK
[6]Faculty of Medical Sciences & Nuffield Department of Orthopaedics Rheumatology and Musculoskeletal Sciences, University of Oxford, Oxford, UK

**Acknowledgements** The authors acknowledge the contributions of other members of the wider OPAL study team. The OPAL investigators acknowledge the support of the British Orthopaedic Association Surgery Research Centre (BOSRC) which supports the development of surgical trials.

**Collaborators** OPAL Collaborators: The following researchers contributed to the concept, design, delivery and analysis of the OPAL study: Sayeed Khan (Make UK, the Manufacturers Organisation), Louise Thomson (Faculty of Medicine and Health Sciences, University of Nottingham), Catherine Hewitt and Catriona McDaid (York Trials Unit, University of York), Iain McNamara (Norfolk and Norwich University Hospital NHS Foundation Trust), David McDonald (University of Stirling), Judith Fitch (British Orthopaedic Association Patient Liaison Group), Gerry Richardson (Centre for Health Economics, University of York). The authors acknowledge the contributions of other members of the wider OPAL study team: Reece Walker and Amanda Goodman (South Tees Hospitals NHS Foundation Trust); Sarah Ronaldson and Elizabeth Coleman (York University); Dr Fiona Nouri and Dr Melanie Narayanasamy (University of Nottingham).

**Contributors** PB: If the chief investigator for the OPAL study. Contributed to the conception of the study, the design of the work; the acquisition, analysis, or interpretation of data for the work; drafted the manuscript. LK: Contributed to the design of the work; the acquisition, analysis or interpretation of data for the work; drafted the manuscript. CC: Contributed to the conception of the study, the design of the work; the interpretation of data for the work; edited the manuscript. AD: Contributed to the conception of the study, the design of the work; the interpretation of data for the work; edited the manuscript. CM: Contributed to the conception of the study, the design of the work; the interpretation of data for the work; edited the manuscript. AR: Contributed to the design of the work; the acquisition, analysis or interpretation of data for the work; drafted the manuscript.

**Funding** This project was funded by the National Institute for Health Research Health Technology Assessment (HTA) programme (project number 15/28/02). Further information available at https://www.journalslibrary.nihr.ac.uk/programmes/hta/152802/#/

**Disclaimer** This report presents independent research commissioned by the National Institute for Health Research (NIHR). The views and opinions expressed by the authors in this publication are those of the authors and do not necessarily reflect those of the National Health Service, the National Institute for Health Research, Medical Research Council, Central Commissioning Facility, NIHR Evaluations, Trials and Studies Coordinating Centre, the Health Technology Assessment Programme or the Department of Health.

**Competing interests** AR is supported by the National Institute for Health Research (NIHR) Oxford Biomedical Research Centre (BRC). CM is a member of the NIHR HTA and EME Editorial Boards.

**Patient and public involvement** Patients and/or the public were involved in the design, or conduct, or reporting, or dissemination plans of this research. Refer to the Methods section for further details.

**Patient consent for publication** Not required.

**Ethics approval** This study was reviewed and given favourable opinion by the East Midlands - Derby Research Ethics Committee (16/EM/0341) and approved by the Health Research Authority in UK.

**Provenance and peer review** Not commissioned; externally peer reviewed.

**Data availability statement** Data are available on reasonable request. Anonymised data detailing the individual participants responses within each round of the OPAL Delphi study is available on request. Please submit requests in writing to the Department of Research and Innovation, South Tees Hospitals NHS Foundation Trust, Marton Rd, Middlesbrough, UK.

**ORCID iDs**
Paul Baker http://orcid.org/0000-0001-8529-2417
Catriona McDaid http://orcid.org/0000-0002-3751-7260
Amar Rangan http://orcid.org/0000-0002-5452-8578

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
