## [Reviewer comments · BMJ Open]

ARTICLE DETAILS

TITLE (PROVISIONAL)	Development of an occupational advice intervention for patients undergoing elective hip and knee replacement: A Delphi study
AUTHORS	Baker, Paul; Kottam, Lucksy; Coole, Carol; Drummond, Avril; McDaid, Catriona; Rangan, Amar

VERSION 1 – REVIEW

REVIEWER	Toby Smith University of East Anglia, UK; University of Oxford, UK; Norfolk and Norwich University Hospitals NHS Foundation Trust, UK
REVIEW RETURNED	08-Jan-2020

GENERAL COMMENTS	Thank you for the opportunity to review this paper. I believe it to be important research which should be published after minor corrects in reporting. These are itemised below. Abstract – this is clear and appropriate. There is repetition in the last lines of the Consensus and Results sections which feels a little redundant. Conclusions appropriate. Could the authors provide more information on the target participants i.e. health professionals, patients, commissioners, employers? This would be helpful to contextualise. Furthermore could they acknowledge whether this was an online Delphi survey and maintained to UK or international respondents? This helps the contextualisation of the work. Strengths and limitations - is 66 participants really a large number of members? I would question this in Point 2. Could a limitation be the generalisability of findings to UK-base? Introduction – The context and background to the work is excellent. There is a strong rationale for why this is important. Exploration on other occupational approaches (if there are) for supporting people to return to work after joint replacement may be helpfully emphasised further. Further information on why a Delphi approach was needed at this stage may have been a useful addition (in 1 or 2 sentences) as opposed to other consensus approaches. Methods – Further clarification for the sample size should be justified based on a theoretical principle. i.e. was this sufficient given the heterogeneity in occupations? Justification is required in the recruitment section. The Delphi approach itself was complex, with adaptive elements between Rounds 1-3 rather than just representing cut-down/augmented options to members. This is articulated clearly within the text. Please provide further details on how reminders were managed i.e. when and how often. What
--

	approaches were made to control NHS/UK/international respondents? Appreciating the sampling/recruitment approach with further clarification would be beneficial. The a priori rules for statement adoption were clearly presented. Further clarification on where the 70% rule was obtained would be helpful for clarification. Results – the results from each round are clearly presented in text form. I wonder whether the use of a figure may help particularly for Round 1 and 2. Ultimately we want to see how the team came to the Intervention in a clear and transparent fashion, but the dedication of this number of words may be rather generous. If this can be distilled into graphical illustration, that may help the reader. Similarly Figure 1 is really helpful. Well done. Would it be possible to provide further information on the respondents and particularly which of the large number of organisations listed in Table 2 contributed. This would aid interpretation. Discussion – the Discussion focussed on defending the limitations of the study. Low sample sizes per stakeholder group is a major issue which is tackled appropriately. However I would recommend the authors consider the issue of generalisability in relation to UK-based findings. Further, consideration on how these findings ‘fit’ within previous understanding on interventions/advice/approaches to encourage and support return to work for people following hip and knee replacement would be helpful. This is touched on but could be developed further. Conclusions – appropriate and I support their recommendation regarding the need to assess this as part of a definitive RCT. Tables and Figures – appropriate Spelling Title Page – inconsistency between universities and departments across the author list – please be consistent. Methods – line 58 – ‘txt’ should be ‘text’
--	--

VERSION 1 – AUTHOR RESPONSE

Reviewer(s)' Comments to Author:

Abstract – this is clear and appropriate. There is repetition in the last lines of the Consensus and Results sections which feels a little redundant.

Response: To avoid repetition we have changed the sentence in the Results section to ‘Eleven respondents provided an appraisal of the intervention in Round 3.’

Conclusions appropriate. Could the authors provide more information on the target participants i.e. health professionals, patients, commissioners, employers? This would be helpful to contextualise.

Response: We have added further information about the target participants of the intervention to the objectives section of the abstract

Furthermore could they acknowledge whether this was an online Delphi survey and maintained to UK or international respondents? This helps the contextualisation of the work.

Response: Reference to this being a UK project is already given in the 'Setting' section of the abstract 'recruited from across the United Kingdom'. This information is also given in the body of the text.

Strengths and limitations - is 66 participants really a large number of members? I would question this in Point 2. Could a limitation be the generalisability of findings to UK-base?

Response: By the standards of many Delphi processes 66 participants is a large number. We therefore stand by this statement. We have added the limitation of this being a UK based process to the strengths and limitations section of the abstract.

Introduction – The context and background to the work is excellent. There is a strong rationale for why this is important. Exploration on other occupational approaches (if there are) for supporting people to return to work after joint replacement may be helpfully emphasised further. Further information on why a Delphi approach was needed at this stage may have been a useful addition (in 1 or 2 sentences) as opposed to other consensus approaches.

Response: Thank you for this comment. We cannot provide an exploration on other occupational advice approaches for hip and knee replacement patients as there is nothing reported in the literature. As part of the OPAL project we undertook a rapid evidence synthesis (PROSPERO protocol registration number CRD42016045235) that found only 4 studies that examined occupational advice interventions in the context of elective surgery (2 RCTs (Belgium and Netherlands) – one in lumbar disc surgery and 1 in gynaecological surgery, 2 qualitative studies (UK and US) – one in arthroplasty patients and 1 in patients post breast surgery). The 1 qualitative study of patient's perceptions of return to work in the orthopaedic setting examined what patients would like to see in a return to work intervention but did not actually describe an approach to delivering RTW advice and support. There is therefore nothing describing an alternative return to intervention in the hip and knee replacement literature. We have added a paragraph to the discussion covering this comment.

We had added in the following sentence to describe why a Delphi consensus process was chosen 'The Delphi approach was chosen as it can be delivered remotely in a short timeframe without the need to convene meetings. It also enables researchers to collect the opinions of a range of different individuals with differing areas of expertise which was desirable in this setting.'

Methods – Further clarification for the sample size should be justified based on a theoretical principle. i.e. was this sufficient given the heterogeneity in occupations? Justification is required in the recruitment section.

Response: Although there are no definitive rules about the sample size for a Delphi study with researchers using 3 to 80 participants, a minimum of 8 to 10 has been suggested. Whilst higher response rates and ease of administration is an advantage with smaller homogeneous groups, it is recommended to give consideration to the expertise of the participants over the actual sample size, specifically if areas of uncertainty are being explored. Larger sample size also reduces errors and improves the reliability of the process. We have used this information and suitable references to explain our rationale for the sample size within the revised manuscript.

Heterogeneity in the occupations and backgrounds of the stakeholder group with a larger sample size offered a wide range of perspectives to identify and address areas of uncertainty which was important to refine the components of a multi-stakeholder intervention within our Delphi study.

We have incorporated some of this text into the manuscript to provide the clarification the reviewer requests.

The Delphi approach itself was complex, with adaptive elements between Rounds 1-3 rather than just re-presenting cut-down/augmented options to members. This is articulated clearly within the text.

Response: We are pleased that this was clear and understandable. Thank you.

Please provide further details on how reminders were managed i.e. when and how often. What approaches were made to control NHS/UK/international respondents? Appreciating the sampling/recruitment approach with further clarification would be beneficial.

Response: All 3 rounds allowed a minimum of 3 weeks for participants to respond. Automated reminders were sent via the electronic system after 10 days from the day of initialising the survey. A further personalised Email reminder was sent to non-responders during the final week of the surveys. This text has been added to the manuscript.

The a priori rules for statement adoption were clearly presented. Further clarification on where the 70% rule was obtained would be helpful for clarification.

Response: There is no universal agreement on an acceptable level of consensus. However, reports suggest this should be decided before commencing the study and recommends at least 70% for validity. We have added in text to explain this within the manuscript.

Results – the results from each round are clearly presented in text form. I wonder whether the use of a figure may help particularly for Round 1 and 2. Ultimately we want to see how the team came to the Intervention in a clear and transparent fashion, but the dedication of this number of words may be

rather generous. If this can be distilled into graphical illustration, that may help the reader. Similarly Figure 1 is really helpful. Well done.

Response: An additional diagram has been added as figure (figure 2) within the manuscript to address this point.

Would it be possible to provide further information on the respondents and particularly which of the large number of organisations listed in Table 2 contributed. This would aid interpretation.

Response: The expression of interest to participate in the Delphi study was publicised via all of the platforms listed in Table 2. However, it was not possible to fully assign an organisational role to a participant as majority of participants contacted the study team directly upon receiving the invite and some organisations provided the contact details for a nominated employer representative. We have provided details of the respondents below. However, we feel that publishing this information in the manuscript may infringe participant confidentiality and therefore have not included it within the revised manuscript.

Patient group respondents included: OPAL Phase 1 Patients'; Patient lead NJR PLG; Service Manager for Arthritis Care; Patient / Ambassador for Global alliance for MSK Health (Formerly the bone and joint decade).

Employers and occupation health services respondents included: Physio manager- Nottingham University; Senior Physiotherapists & Occupational Health and Training Team Managers from Rhondda Cynon Taf Council; Head of Safety, Health and Quality from Finning UK & Ireland; Occupational Health advisor from Briar Chemicals, Norwich.

Orthopaedic surgeons: Research lead BASK; Past-president BASK; Past president BHS & other consultant orthopaedic Surgeons

AHPs:

Physios from Middlesbrough, Norwich, Bournemouth

OTs from Derby Darlington, Norwich, St Helens & Tayside NHS

Nurse practitioners and Joint replacement nurses from Edinburgh & Middlesbrough NHS

GPs:

GPs from Edinburgh, Leicestershire, Northumberland; Academic GP from Liverpool; RCGP lead for chronic pain (currently in clinical Research) & Occupational Health Physician from Manchester

Discussion – the Discussion focussed on defending the limitations of the study. Low sample sizes per stakeholder group is a major issue which is tackled appropriately. However I would recommend the authors consider the issue of generalisability in relation to UK-based findings.

Response: We have added in a sentence about the generalisability of the findings and the developed intervention outside of the UK to the discussion

Further, consideration on how these findings ‘fit’ within previous understanding on interventions/advice/approaches to encourage and support return to work for people following hip and knee replacement would be helpful. This is touched on but could be developed further.

Response: As per earlier comments there are no published reports of interventions for hip and knee replacement patients. We have now specifically described this issue in the discussion and referenced the findings of the OPAL rapid evidence synthesis described earlier.

Conclusions – appropriate and I support their recommendation regarding the need to assess this as part of a definitive RCT.

Response: Thank you for this comment. No specific action required

Tables and Figures – appropriate

Response: Thanks you. No action required

Spelling

Title Page – inconsistency between universities and departments across the author list – please be consistent.

Response: We have reviewed the affiliations and updated them to make them more consistent

Methods – line 58 – ‘txt’ should be ‘text’

Response: This has been corrected

VERSION 2 – REVIEW

REVIEWER	Toby Smith University of East Anglia & University of Oxford
REVIEW RETURNED	16-Apr-2020

GENERAL COMMENTS

The authors have addressed my original concerns. The paper is a valuable contribution to the evidence-base. I strongly support it's current publication as it stands. Well done.